# Uncertainty-Driven Pessimistic Q-Ensemble for Offline-to-Online Reinforcement Learning

**Ingook Jang**
Electronics and Telecommunications Research Institute
ingook@etri.re.kr

**Seonghyun Kim**
Electronics and Telecommunications Research Institute
kim-sh@etri.re.kr

## Abstract

Re-using existing offline reinforcement learning (RL) agents is an emerging topic for reducing the dominant computational cost for exploration in many settings. To effectively fine-tune the pre-trained offline policies, both offline samples and online interactions may be leveraged. In this paper, we propose the idea of incorporating a pessimistic Q-ensemble and an uncertainty quantification technique to effectively fine-tune offline agents. To stabilize online Q-function estimates during fine-tuning, the proposed method uses uncertainty estimation as a penalization for a replay buffer with a mixture of online interactions from the ensemble agent and offline samples from the behavioral policies. In various robotic tasks on D4RL benchmark, we show that our method outperforms the state-of-the-art algorithms in terms of the average return and the sample efficiency.

## 1 Motivation

Deep offline reinforcement learning (RL) [9] trains deep neural networks from previously collected datasets to learn powerful robotic agents without additional environmental interactions. Recently, offline RL methods [4, 6, 1, 11, 7] have performed better than the behavioral policies which produce the offline datasets. Offline RL agents, however, may perform suboptimally 1) when the offline datasets follow possibly suboptimal behavioral policies; 2) when the behavioral policies are insufficiently explorative; or 3) when the datasets are too small. This requires further online learning (fine-tuning) from additional online interactions generated by offline RL agents. The goal of online fine-tuning of offline agents is to improve sample efficiency and asymptotic performance of online learning by efficiently leveraging offline datasets.

To fine-tune an offline RL agent, slightly modified off-policy RL methods may be used by leveraging both online and offline samples in training. However, such modifications are typically difficult to make learning sample-efficient or asymptotically optimal due to distributional shift. Fine-tuning an agent trained solely on an offline dataset may yield inaccurate value estimates due to online visitation of out-of-distribution (OOD) state-action samples. The distributional shift between the offline dataset and observed online interactions causes large initial temporal difference errors, which in turn causes initial performance degradation and the agent to forget all information obtained from offline RL. This leads to decreased sample efficiency by losing the benefit of having an initial good policy.

To address this problem, offline-to-online RL (Off2OnRL [8]) leverages a pessimistic Q-ensemble scheme. Off2OnRL pessimistically trains multiple Q-functions to mitigate such bootstrapping errors caused by the distributional shift. This constrains the learning policy to visit near the distribution

Offline Reinforcement Learning Workshop at Neural Information Processing Systems, 2022.

of the behavioral policy to stabilize fine-tuning an offline RL agent. Off2OnRL also introduced a prioritized buffer with balanced replay, which contains not only online interactions collected during fine-tuning but near-on-policy samples from the offline dataset. This scheme enables accurate Q-function estimation by training a network for prioritizing offline samples, which can bring the sampling distribution for Q-functions close to online samples.

The goals of online learning using an offline dataset are 1) to estimate a good initial policy that is not immediately forgotten during online learning; and 2) to improve sample efficiency. However, we find that online fine-tuning incorporated with the offline dataset tends to be affected by uncertainty estimation since the buffer has online and offline samples generated from different state-action distributions. It may suffer from high uncertainty when 1) the network for the balanced replay is not trained sufficiently; and 2) the offline dataset tends to be diverse around several trajectories produced by multiple policies.

In this paper, we introduce an uncertainty-driven pessimistic Q-ensemble (UPQ) for offline-to-online RL. We adopt a pessimistic ensemble of offline actor-critic agents to guide the learning policy with efficient pessimism. To stabilize online Q-function estimates during online fine-tuning, the proposed method leverages uncertainty quantification as a penalization for a prioritized replay buffer where online samples from the ensemble agent and offline samples from the behavioral policy exist together. In our experiments, we demonstrate that our method obtains high-performing policies with fewer online interactions, and hence outperforms the state-of-the-art algorithms in terms of the average return and the sample efficiency.

## 2 Methodology

We consider a Markov Decision Process (MDP) tuple $\mathcal{G} = (\mathcal{S}, \mathcal{A}, r, \mathbb{P})$ defined by the state space $\mathcal{S}$, the action space $\mathcal{A}$, the reward function $r$, and the transition distribution $\mathbb{P}$. The objective of an RL agent is to maximize the expected return $\mathbb{E}[\sum_{i=0}^{T-1} \gamma^i r_i]$, where $\gamma \in [0, 1)$ is the discount factor and $T$ is the episode horizon (total number of timesteps of an episode) that the agent optimizes over.

In our proposed method, we use $N$ offline actor-critic agents $\{Q_{\theta_i}, \pi_{\phi_i}\}_{i \in [N]}$ for an ensemble from Off2OnRL [8], where $\theta_i$ and $\phi_i$ are defined as the parameters of the $i$ agent's critic and actor, respectively. We define the Q-function and the policy of the ensemble as follows:

$$Q_\theta^E(s, a) := \frac{1}{N} \sum_{i=1}^{N} Q_{\theta_i}(s, a), \tag{1}$$

$$\pi_\phi^E(\cdot|s) = \mathcal{N}\left(\frac{1}{N} \sum_{i=1}^{N} \mu_{\phi_i}(s), \quad \frac{1}{N} \sum_{i=1}^{N} \left(\sigma_{\phi_i}^2(s) + \mu_{\phi_i}^2(s)\right) - \mu_\phi^2(s)\right), \tag{2}$$

where the parameters are defined as $\theta := \{\theta_i\}_{i \in [N]}$ and $\phi := \{\phi_i\}_{i \in [N]}$, respectively. The defined $\pi_\phi^E$ follows a normal distribution with mean and variance of the Gaussian mixture $\frac{1}{N} \sum_{i=1}^{N} \pi_{\phi_i}$ for parameterization. The modeled policy is the same as Off2OnRL [8].

Since the ensemble estimates the posterior distribution of its Q-functions, we use the standard deviation-based uncertainty quantification technique proposed in Pessimistic Bootstrapping for offline RL (PBRL [2]). The uncertainty estimation at $(s', a')$ of the target Q-functions is defined as follows:

$$\mathcal{U}_{\theta-}(s', a') := \sigma(Q_{\theta_{i-}}(s', a')) = \sqrt{\frac{1}{N} \sum_{i=1}^{N} \left(Q_{\theta_{i-}}(s', a') - Q_{\theta-}^E(s', a')\right)^2}, \tag{3}$$

where $\theta_{i-}$ is the parameters of the $i$ agent's target Q-network and we denote the mean over the ensemble of the target Q-functions by $Q_{\theta-}^E$. In policy evaluation, we use such uncertainty quantification as a penalization to the next Q-value for a mixture of online and offline samples from the prioritized replay buffer $\mathcal{B}$ proposed in Off2OnRL. The $Q^E$ of the ensemble agent is updated through pessimistic Q-function updates by fitting the following target for state-action pairs sampled from $\mathcal{B}$:

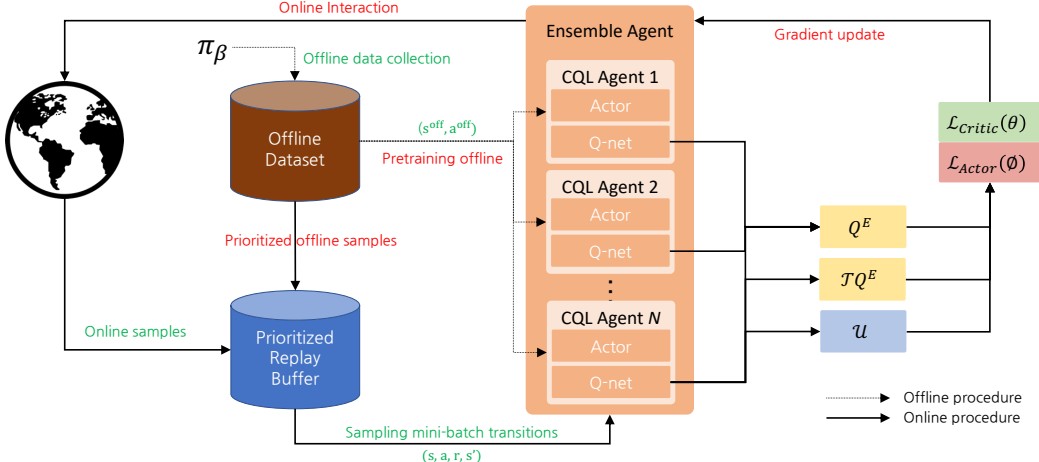

Figure 1: Illustration of the overall architecture of the proposed method with a pessimistic Q-ensemble and uncertainty quantification. Our method calculates the Q-target $\mathcal{T}Q^E$ and the corresponding uncertainty $\mathcal{U}$ by using the ensemble of the $N$ offline agents pre-trained from the offline dataset to update the gradients.

$$\mathcal{T}Q_\theta^E(s,a) := r(s,a) + \gamma \mathbb{E}_{a' \sim \pi_\phi^E} \left[ Q_{\theta-}^E(s',a') - \alpha \log \pi_\phi^E(a'|s') - \beta \mathcal{U}_{\theta-}(s',a') \right], \qquad (4)$$

where $\beta$ is the tuning parameter for the uncertainty penalization. To this end, the parameters of the Q-network and the policy of the ensemble agent, $\theta$ and $\phi$, are updated by minimizing the following objectives, respectively:

$$\mathcal{L}_{Critic}(\theta) = \mathbb{E}_{(s,a,s') \sim \mathcal{B}} \left[ \left( Q_\theta^E(s,a) - \mathcal{T}Q_\theta^E(s,a) \right)^2 \right], \qquad (5)$$

$$\mathcal{L}_{Actor}(\phi) = \mathbb{E}_{s \sim \mathcal{B}, a \sim \pi_\phi^E} \left[ \alpha \log \pi_\phi^E(a|s) - Q_\theta^E(s,a) \right], \qquad (6)$$

where $\alpha$ is the parameter for temperature.

Figure 1 illustrates the overall workflow of the proposed method with uncertainty quantification. The learning Q-function and policy are updated via update rules (5) and (6) by fine-tuning the pre-trained multiple ensemble agents.

## 3  Experimental Setup and Results

We use the D4RL benchmark [3], which includes datasets for data-driven deep reinforcement learning. To demonstrate the performance of our method, three Mujoco [10] locomotion tasks (HalfCheetah, Walker2d, and Hopper) are adopted with four types of datasets (random, medium, medium-replay, medium-expert) for comparative evaluation. Specifically, the random dataset has collections by a random policy and the medium dataset contains samples from a medium-level policy trained via Soft Actor-Critic (SAC [5]). The medium-replay dataset contains all samples observed during training a medium-level policy and the medium-expert dataset consists of samples recorded by both a medium-level agent and expert demonstrations. We use a total of 12 task setups with the 'v2' version datasets.

We compare the proposed method with several baseline algorithms including CQL [7] and Off2OnRL [8]. CQL provides a good baseline for the performance of an offline approach, which is also used for learning both Off2OnRL and our method. The implementation of CQL and Off2OnRL are adopted from the official implementation at `https://github.com/aviralkumar2907/CQL` and `https://github.com/shlee94/Off2OnRL`, respectively. We fix most of the hyperparameters for

Table 1: Average returns for all algorithms in Mujoco locomotion tasks. The highest average return values are highlighted.

| | | CQL | Off2onRL | UPQ $\beta = 0.1$ | UPQ $\beta = 0.01$ | UPQ $\beta = 0.001$ |
|---|---|---|---|---|---|---|
| **Random** | HalfCheetah | 2455.2 | 11474.5 | 11068.6 | 11172.4 | 11280.5 |
| | Hopper | 323.5 | 3213 | 945.3 | 2622.6 | 2967.8 |
| | Walker2d | 372.2 | 2706.1 | 2224.1 | 3190.8 | 2759.2 |
| **Medium** | HalfCheetah | 5171.1 | 10049.7 | 10364.9 | 10847.9 | 10668.2 |
| | Hopper | 1973.4 | 3279.7 | 3416.8 | 3361.3 | 3424.1 |
| | Walker2d | 3288.4 | 4638.5 | 4356.5 | 4736.9 | 4971.2 |
| **Medium Replay** | HalfCheetah | 5214 | 10413.4 | 10118.3 | 10600.4 | 10313.2 |
| | Hopper | 1698.8 | 3521.6 | 3195.2 | 3206.7 | 3440.4 |
| | Walker2d | 3142.1 | 4760.9 | 4505.2 | 4750.9 | 5100.2 |
| **Medium Expert** | HalfCheetah | 1158.8 | 11159.0 | 11261.2 | 11277.9 | 11353.4 |
| | Hopper | 1272.1 | 2531.3 | 2201.1 | 2594.8 | 3423.4 |
| | Walker2d | 3675 | 5181.8 | 5525.7 | 5682.6 | 5446 |

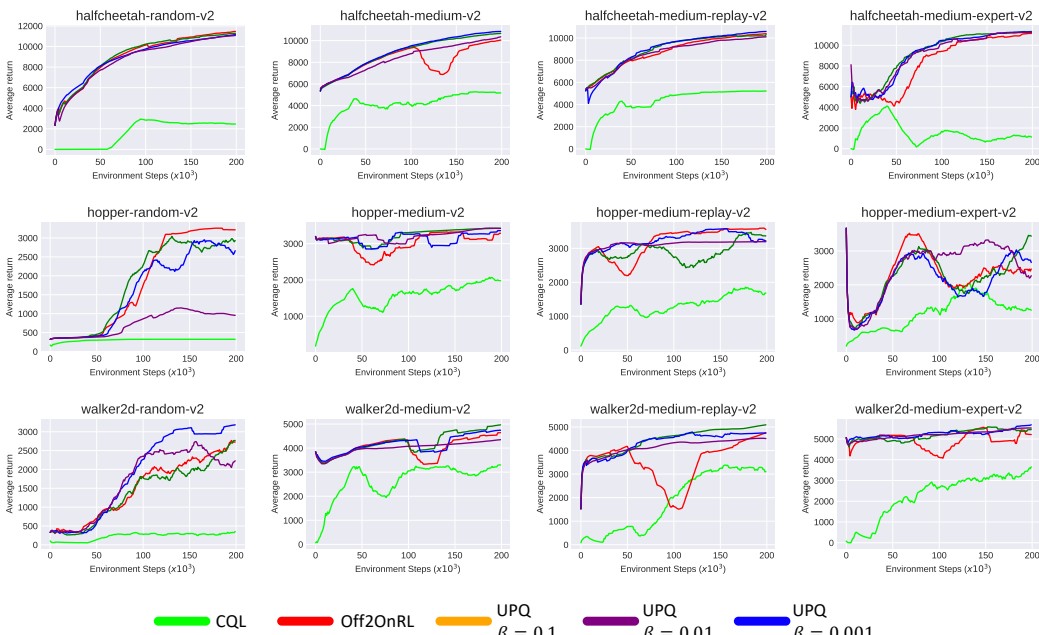

Figure 2: Training curves of all algorithms in Mujoco locomotion tasks.

our method the same as the official Off2OnRL implementation. We train $N = 4$ CQL agents for 1000 epochs with different seeds for all the locomotion tasks where we experiment. Off2onRL and our method are trained for 200 epochs (200000 environmental steps) using the pre-trained CQL agents. In our method, the parameter for uncertainty penalization $\beta$ is varied from $[0.1, 0.01, 0.001]$.

Table 1 reports the performance of our method and baseline algorithms for each task. UPQ performs similarly or better than the baseline methods in most of the tasks in terms of the average return and sample efficiency. We find that UPQ has advantages in the tasks with non-optimal datasets marked as medium, medium-replay, and medium-expert. The experiments show that UPQ performs the best with $\beta \in [0.01, 0.001]$ in many tasks. Figure 2 illustrates the training curves for UPQ and baseline algorithms in all tasks. We remark that the performance of UPQ shows more stable training curves in the early stage (especially $50000 - 100000$ environmental steps) of fine-tuning in most of the tasks.

# 4  Conclusion

In this work, we propose UPQ with an uncertainty-driven pessimistic Q-ensemble and it is effective for improving offline-to-online RL methods. To stabilize online Q-function estimates during fine-tuning, the proposed method leverages uncertainty quantification as a penalization for a prioritized replay buffer consisting of both online and offline samples. Our experiments show that UPQ outperforms several baselines over robotic locomotion tasks in terms of the average return and the sample efficiency. We expect that our method enables sample-efficient re-use of the existing offline agents and offline datasets. We also expect to implement further improvements as future work.

## Acknowledgments and Disclosure of Funding

This work was supported by Electronics and Telecommunications Research Institute (ETRI) grant funded by the Korean government. [22ZR1100, A Study of Hyper Connected Thinking Internet Technology by autonomous connecting, controlling and evolving ways].

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
