# OpenReview forum: "Uncertainty-Driven Pessimistic Q-Ensemble for Offline-to-Online Reinforcement Learning"
_NeurIPS.cc/2022/Workshop/Offline_RL — Offline RL Workshop NeurIPS 2022_

### Official Review · Reviewer_htar · 2022-10-18
**Review for "Uncertainty-Driven Pessimistic Q-Ensemble for Offline-to-Online Reinforcement Learning"**

**Rating:** 6
**Confidence:** 3

**Review:**

OVERVIEW:

Traditional reinforcement learning (RL) algorithms have limited applicability in many real-world situations where repeated large-scale data collection is costly or dangerous. In contrast, offline RL can learn from large existing data banks of previously-logged interactions without requiring additional data collection. The problem with offline RL agents is that their performance is usually inferior to that obtained by state-of-the-art online RL algorithms. Therefore, *offline-to-online RL*, or the ability to leverage offline datasets to reduce exploration cost and improve sample efficiency for online RL, is an important area of application. To this end, the authors improve upon the state-of-the-art in offline-to-online RL and appear to achieve even better performance on common locomotion benchmarks. However, while I feel this work has potential to be a worthwhile contribution to the literature, I also have several serious critiques with the paper as it currently stands.

PROS:

1. You propose a novel extension to a state-of-the-art method for offline-to-online learning that takes into account estimation uncertainty.

2. Your extension is intuitively reasonable and follows logically from previous literature cited in your paper.

3. You assessed the performance of your method on several common and challenging benchmarks for deep offline RL.
$\quad\quad\quad\quad\quad\quad\quad\quad\quad\quad\quad\quad\quad\quad\quad\quad\quad\quad\quad\quad\quad\quad\quad\quad\quad\quad\quad\quad\quad\quad\quad\quad\quad\quad\quad\quad\quad\quad\quad\quad\quad\quad\quad\quad\quad$


CRITICAL ISSUES:


1. **Readers will not understand your paper unless they have read the Off2OnRL paper first.** It should be clearly elaborated what the goal of offline-to-online RL or online fine-tuning is: leveraging offline datasets to improve sample efficiency and perhaps asymptotic performance of online RL. Specifically, Off2OnRL (Lee et. al. 2022) and uncertainty-driven pessimistic Q-ensemble (UPQ) seeks to do two things: (1) use an offline dataset to estimate a good initial policy that is not immediately forgotten during online learning; and (2) further incorporate the offline dataset in the replay buffer during online learning to improve sample efficiency. Doing both effectively will allow us to obtain a high-performing policy with less online interactions than just pure online RL. The logic behind offline-to-online RL was elaborated only briefly in the first paragraph in the "Motivation" section, and your elaboration needs to be expanded to more clearly explain the theoretical advantage of offline-to-online RL over applying just pure online RL. You also do not clearly explain why we cannot just use traditionally off-policy methods to apply online learning and improve performance after using offline RL to estimate an initial policy. This is explained only briefly in the second paragraph of the "Motivation" section, and your explanation needs to be expanded. For example, it should be explained that the distributional shift between the offline dataset and observed online interactions causes large initial temporal difference errors, which in turn causes initial performance degradation and the agent to forget all information obtained from offline RL. By losing the benefit of having an initial good policy, we have decreased sample efficiency.
$\quad\quad\quad\quad\quad\quad\quad\quad\quad\quad\quad\quad\quad\quad\quad\quad\quad\quad\quad\quad\quad\quad\quad\quad\quad\quad\quad\quad\quad\quad\quad\quad\quad\quad\quad\quad\quad\quad\quad\quad\quad\quad\quad\quad\quad\quad\quad\quad\quad\quad\quad\quad\quad\quad\quad\quad\quad$
Finally, the "Methodology" section does not go through the methodological details of UPQ that are borrowed from Off2OnRL. For example, there was no mention on how the ensemble of agents (actors and critics) were created. It should be explained that each agent was trained independently via conservative Q-learning (CQL), each with different initial parameters and a different order of mini-batches visited during training. The methodological details of the prioritized replay buffer used to combine the offline dataset and online replay buffer should also be explained. Lastly, you should elaborate in more detail as to why an ensemble of pessimistic Q-functions and prioritized replay are critical for effective offline-to-online RL. I understand that Lee et. al. 2022 discusses many of these points already, and that discussing these issues in-depth may have been difficult due to the page limit of the submission paper. However, for the camera-ready version, your paper should be more self-contained so that readers who have not read Lee et. al. 2022 can still understand your paper.
$\quad\quad\quad\quad\quad\quad\quad\quad\quad\quad\quad\quad\quad\quad\quad\quad\quad\quad\quad\quad\quad\quad\quad\quad\quad\quad\quad\quad\quad\quad\quad\quad\quad\quad\quad\quad\quad\quad\quad\quad\quad\quad\quad\quad\quad$

2. The last two paragraphs of the "Motivation" section state that the balanced replay method tends to be affected by uncertainty estimation, and that you propose uncertainty-driven pessimism to this end. However, **it is not clear why and how the balanced replay method makes uncertainty regularization necessary.** For example, it was stated in the second-to-last paragraph of the "Motivation" section that the balanced replay method suffers from high uncertainty when "the network for the balanced replay is not trained sufficiently". By "the network", do you mean the density ratio estimator used for prioritized replay? If the density ratio estimator is inaccurate, why do you propose adjusting for estimation uncertainty in the critics and not the density ratio estimator? The same paragraph also states that uncertainty is an issue when "the offline dataset tends to be diverse around several trajectories produced by multiple policies". How does offline dataset diversity have anything to do with balanced replay or critic estimation uncertainty? I understand how uncertainty regularization could improve performance in offline and online RL more generally, but I don't understand how this has anything to do with offline-to-online RL or balanced replay specifically. I think it would be helpful to conduct some experiments (whether it be on simple tabular environments or MuJoCo environments) where you show empirically that estimation uncertainty increases in scenarios where the density ratio estimator is not fully trained or where the offline dataset contains trajectories from multiple diverse policies. You can then use the same experiments to demonstrate that your method reduces estimation uncertainty and improves performance in these scenarios. Such experiments would better communicate to readers the issues with Off2OnRL and better justify your proposed extension.
$\quad\quad\quad\quad\quad\quad\quad\quad\quad\quad\quad\quad\quad\quad\quad\quad\quad\quad\quad\quad\quad\quad\quad\quad\quad\quad\quad\quad\quad\quad\quad\quad\quad\quad\quad\quad\quad\quad\quad\quad\quad\quad\quad\quad\quad$

3. **Your measure of estimation uncertainty may be suboptimal**. Assuming you trained the offline agents in your ensemble identically to Off2OnRL, each agent would have been trained on the same data, albeit with differences in the parameter initialization and mini-batch sampling. Then based on equations 5 and 6, it appears that you use the same replay buffer, mini-batches and targets to update each network in the critic ensemble during online fine-tuning. This means that only the initialization and optimizer varies between your critic networks for offline learning, and only the initialization varies between your critics for online learning. In contrast, each critic network in a bootstrapped DQN ensemble (Osband et. al. 2016) uses a different initialization, target network and replay buffer during online learning. This difference matters because only an ensemble from bootstrapped DQN accounts for stochasticity in the optimizer and data-generating process and would truly capture estimation uncertainty. Recall that your technique for measuring estimation uncertainty was adopted from Bai et. al. 2022, and that the ensemble technique used in Bai et. al. 2022 was itself adopted from bootstrapped DQN.  Another issue is that your ensemble consists of only four Q-functions, while other implementations that account for estimation uncertainty often involve larger ensembles.
$\quad\quad\quad\quad\quad\quad\quad\quad\quad\quad\quad\quad\quad\quad\quad\quad\quad\quad\quad\quad\quad\quad\quad\quad\quad\quad\quad\quad\quad\quad\quad\quad\quad\quad\quad\quad\quad\quad\quad\quad\quad\quad\quad\quad\quad\quad\quad\quad\quad\quad\quad\quad\quad\quad\quad\quad\quad$
With these points in mind, I would recommend conducting a few ablation studies. First, I would recommend exploring more than four CQL agents: While four CQL agents may have been sufficient for Off2OnRL, a larger ensemble may be needed for accurate uncertainty quantification. Second, I would try diversifying how each critic network is trained, particularly during online learning. This can be done as in Osband et. al. 2016 whereby each critic uses its own target network. Alternatively, each critic could use the same target network but use different mini-batches for applying updates. Moreover, while Osband et. al. found that perturbing the replay buffer did not yield improved performance over no perturbations, they only used the estimation uncertainty in their ensemble implicitly for exploration, while you are trying to quantify estimation uncertainty explicitly. Perturbing the replay buffer that each critic network is trained on during offline and online learning as explored in Osband et. al. 2016 (see section B of the Appendix as well as section 6 of the main paper) may improve performance due to improved uncertainty quantification as well as reduced estimation variance of the ensemble average (Hastie, Tibshirani and Friedman 2009).
$\quad\quad\quad\quad\quad\quad\quad\quad\quad\quad\quad\quad\quad\quad\quad\quad\quad\quad\quad\quad\quad\quad\quad\quad\quad\quad\quad\quad\quad\quad\quad\quad\quad\quad\quad\quad\quad\quad\quad\quad\quad\quad\quad\quad\quad$

4. **It is difficult to infer the results of your experiments and the relative performance of your method.** Table 1 is difficult to interpret because it is comparing UPQ to competitors while simultaneously comparing UPQ for different choices of $\beta$. Figure 2 is even more difficult to interpret: with a different graph per dataset and three versions of UPQ per graph, it is very difficult to see if and how UPQ improves upon Off2OnRL. I think the performance gains of UPQ over its competitors would be communicated more clearly if only one value of $\beta$ was chosen for each dataset for the purposes of Table 1 and Figure 2. You can either do this by using the same value of $\beta$ for every dataset, or by adapting the value of $\beta$ to each dataset. If you choose the latter approach, the procedure you use to tune $\beta$ should be one requiring only minimal additional online interaction. You can then discuss how performance changes with different choices of $\beta$ in a separate table. Summarizing results across plots in Figure 2 via some kind of quantitative metric, such as mean area under the curve (AUC), could also help communicate results (Stadie, Levine and Abbeel 2015). Finally, I think more baselines should be explored. Several baselines explored in Lee et. al. 2022 combined offline RL and online fine-tuning and could serve as a useful reference for this purpose. It would also be a good idea to include baselines that utilize uncertainty quantification during online updates, assuming such baselines have been proposed in previous literature.


Other issues:

1. On line 13, you state "Deep offline reinforcement learning (RL) utilizes deep neural networks trained from previously collected offline datasets to learn..." Offline RL doesn't actually use trained networks to learn, but rather it learns by training. This should be re-written as "deep offline reinforcement learning (RL) trains deep neural networks from previously collected datasets to learn..."

2. On line 16 you state "Offline RL agents, however, may perform suboptimally when the offline datasets follow possibly suboptimal behavioral policies". While this is one reason why the offline dataset and offline RL agent may be suboptimal, there may be other reasons, such as the dataset being too small or the behavioral policy being insufficiently explorative.

3. On line 20 you state that when applying traditional off-policy methods to perform online fine-tuning, it "is typically difficult to improve performance due to distributional shift". As shown in Figures 1 and 3 of Lee et. al. 2022, the issue is not that performance won't be improved at all, but rather that learning won't be sample efficient or asymptotically optimal.

3. On line 46 you should change the episode horizon definition from "total number of an episode" to "total number of time steps of an episode". You also never defined the $\theta_{i-}$ term present in equation 3.

4. On line 85 you state "Off2onRL and our method are trained for 200 epochs using the pre-trained CQL agents." Based on Figure 2, I believe you meant to say that they were trained for 200 ITERATIONS, where each iteration consists of 1000 environment steps.

5. There are linguistic errors throughout the paper. For example, on line 15 you write "Recently, offline RL methods often perform better..." when you should write "Recently, offline RL methods have performed better..". On line 20, the part "such modification" should be changed to "such a modification" or "such modifications". The "Experimental Setup and Result" section should be renamed as "Experimental Setup and Results".

$\quad\quad\quad\quad\quad\quad\quad\quad\quad\quad\quad\quad\quad\quad\quad\quad\quad\quad\quad\quad\quad\quad\quad\quad\quad\quad\quad\quad\quad\quad\quad\quad\quad\quad\quad\quad\quad\quad\quad\quad\quad\quad\quad\quad\quad$


REFERENCES:

(1) Seunghyun Lee, Younggyo Seo, Kimin Lee, Pieter Abbeel, and Jinwoo Shin. Offline-to-online reinforcement learning via balanced replay and pessimistic q-ensemble. In Conference on Robot Learning, pages 1702–1712, 2022.

(2) Trevor Hastie, Robert Tibshirani and Jerome Friedman. Elements of Statistical Learning, second edition. Springer New York Inc, 2022.

(3) A. W. van der Vaart. Asymptotic Statistics. Cambridge University Press, 2012.

(4) I. Osband, C. Blundell, A. Pritzel, and B. Van Roy. Deep exploration via bootstrapped dqn. In Advances in Neural Information Processing Systems, 2016.

(5) Chenjia Bai, Lingxiao Wang, Zhuoran Yang, Zhi-Hong Deng, Animesh Garg, Peng Liu, and Zhaoran Wang. Pessimistic bootstrapping for uncertainty-driven offline reinforcement learning. In International Conference on Learning Representations, 2022.

(6) Bradly C Stadie, Sergey Levine, and Pieter Abbeel. Incentivizing exploration in reinforcement learning with deep predictive models. arXiv preprint arXiv:1507.00814, 2015.

---

### Official Review · Reviewer_eryf · 2022-10-19
**Interesting and worth exploring, but could be more rigorous**

**Rating:** 6
**Confidence:** 3

**Review:**

This work propose using uncertainty as a penalty for pessimistic Q-ensemble to fine-tune offline agents.

Questions:
0. In the D4RL benchmark (Table 1), could you report normalized score similar to the Table 1 in CQL paper?
1. Since the work is on uncertainty, have you considered works like as baseline [1,2]?
2. Do the results in Table 1 match results in Table 1 of CQL paper? I thought CQL should perform much better even without online fine-tuning.
3. Have you tried other d4rl benchmarks?

[1] Wu, Yue, et al. "Uncertainty weighted actor-critic for offline reinforcement learning." arXiv preprint arXiv:2105.08140 (2021).
[2] Bai, Chenjia, et al. "Pessimistic bootstrapping for uncertainty-driven offline reinforcement learning." arXiv preprint arXiv:2202.11566 (2022).

Reason for decision:
I think the direction of the study is interesting and worth exploring, but the paper would require more work if submitting to a conference.